# Dynamics of Non-Structural Carbohydrates Release in Chinese Fir Topsoil and Canopy Litter at Different Altitudes

**DOI:** 10.3390/plants12040729

**Published:** 2023-02-07

**Authors:** Xiaojian Wu, Yue Cao, Yu Jiang, Mingxu Chen, Huiguang Zhang, Pengfei Wu, Xiangqing Ma

**Affiliations:** 1College of Forestry, Fujian Agriculture and Forestry University, Fuzhou 350002, China; 2Wuyishan National Park Scientific Research Monitoring Center, Wuyishan 354300, China

**Keywords:** Chinese fir, topsoil litter, canopy litter, non-structural carbohydrates, climate change

## Abstract

Non-structural carbohydrates (NSCs) are labile components in forest litter that can be released quickly at the early stage of litter decomposition and accelerate the metabolic turnover of soil microorganisms, which is essential for the formation of forest soil organic matter. Therefore, understanding the NSCs response mechanisms to forest litter at different altitudes is critical for understanding nutrient cycling in the forest soil under climate change conditions. In this study, we used the net bag decomposition method to observe the dynamics of NSCs release in Chinese fir topsoil and canopy litter at four altitudes for 360 days based on the climatic zone characteristics distributed vertically along the elevation of Wuyi Mountain. The release of NSCs in Chinese fir litter rise gradually with height increases during the decomposition. The difference of the cumulative release percentage of soluble sugar between different altitudes is more significant than that of starch. The response of the NSC content in different treatment groups at four altitudes are different. The release of NSCs in the leaf canopy litter is higher than that in the leaf topsoil litter. On the contrary, the release of NSCs in the mixture of leaf and twig topsoil litter is higher than that in the mixture of leaf and twig canopy litter. Taken together, this study is of great significance for a comprehensive understanding of the effect of climate change on NSCs during the decomposition of Chinese fir litter.

## 1. Introduction

The forest ecosystem is the largest nutrient cycling pool on land, and forest litter is an important carrier of energy flow and nutrient return in forest ecosystems [1]. Nutrients returned by forest litter decomposition can meet 69–87% of the nutrients required for forest growth, which plays an important role in maintaining the long-term productivity of forests [2]. Nutrient release during litter decomposition has an important effect on plant nutrient uptake and the carbon cycle. Raich et al. [3] reported that the CO_2_ released by the decomposition of forest litter accounts for approximately 70% of the global carbon flux and is an important part of the carbon cycle in forest ecosystems. Non-structural carbohydrates (NSCs), a readily decomposable component of forest litter, are extremely sensitive to environmental and climate change [4]. In the early stages of forest litter decomposition, the rapid release of NSCs can provide sufficient energy for soil microorganisms, promote microbial metabolic turnover, and accelerate the nutrient release of forest litter [5]. Therefore, insights into the dynamics of NSCs release during the decomposition of forest litter are important for a comprehensive understanding of forest litter decomposition.

Since the 1950s, climate change has become a cause for concern. IPCC AR6 reported that the global average annual temperature in the next 20 years will be 0.3–0.7 °C higher than that in 2001–2020 [6]. This continuous increase in global temperature not only alters the growth strategy and distribution pattern of vegetation but also disrupts the balance between aboveground and underground carbon pools, which has a profound impact on the productivity and ecological function of forest ecosystems [7]. Wuyi Mountain has the largest and most completely preserved evergreen broad-leaved forest ecosystem at 27° N, with a complete spectral distribution of vegetation along altitudinal belts and distinct distribution characteristics of vertical climate zones [8]. Topography leads to different hydrothermal conditions (temperature, humidity, ultraviolet, etc.), which makes the litter decomposition process vary with altitudes. Salinas et al. [9] found that temperature was the most important factor in explaining the decomposition of forest litter along an altitudinal gradient and that soil temperature could explain 95% of the variation in the annual decomposition rate of forest litter. In recent years, studies on the response of forest litter to climate change have mostly focused on mass loss, nutrient return, and microbial community composition [10,11]. However, the dynamics of NSCs release during litter decomposition remains unresolved.

Chinese fir (*Cunninghamia lanceolata* (Lamb.) Hook.), a unique quick-growing timber species that exhibits fast growth, high yield, high disease resistance, and good material quality, is widely distributed in 17 provinces in China [12]. Studies have found that the biggest difference between Chinese fir and other trees is that a large number of dead leaves and branches remain attached to the trunk for more than four years before abscission [13], and the leaves are brown or greyish-brown in color (“canopy litter”). Previous studies reported that, in 10-year-old Chinese firs, the mass of dead leaves and branches in the canopy accounted for more than 95% of the total mass of litter on the ground and remained at 20.1–25.3% in 15-year-old Chinese firs [13]. Several studies have demonstrated that the canopy litter is not only a component of the nutrient cycling but also plays an important role in the aboveground coarse woody detritus pool [14], and the decomposition rate of canopy litter is much slower than that of topsoil litter [15]. It did not fall to the ground as quickly, which substantially delayed the nutrient return of the dead branches and leaves in Chinese fir plantations. Previous studies on Chinese fir canopy litter have focused on the quantity, nutrient content, and spatial distribution characteristics [16], but the differences between the decomposition processes of topsoil litter and canopy litter of Chinese fir and their responses to climate change are not well known.

We hypothesized that: (1) the release of NSCs in Chinese fir litter show a decline trend with increasing of altitude; (2) the release of NSCs in Chinese fir canopy litter is higher than that in the topsoil litter. The specific objectives of this study were to observe the decomposition process of Chinese fir dead leaves and branches at different altitudes for 360 days, revealing the dynamics of NSCs release during decomposition process of Chinese fir topsoil litter and canopy litter under the background of global climate change.

## 2. Results

### 2.1. Dynamic Changes of Soluble Sugars at Different Altitudes

With the increasing of decomposition time, the cumulative release of soluble sugar in Chinese fir topsoil litter and canopy litter increased at different altitudes (Figure 1). The release rate of soluble sugar in Chinese fir topsoil litter and canopy litter during 0–180 days was higher than that during 240–360 days. During the decomposition period of 120–360 days, the rank order of cumulative release of soluble sugar in the LTL and LTTL groups at four altitudes was as follows: 620 m > 1003 m > 1410 m > 1968 m. During the decomposition period of 180–360 days, the release of soluble sugar in the LCL was highest at 1410 m and lowest at 1968 m. On day 300 of decomposition, the cumulative release percentage of soluble sugar in the LTCL at four altitudes gradually stabilized. On day 360 of decomposition, we observed that the cumulative release percentage of soluble sugar among different treatments are different. The cumulative release percentage of soluble sugar in the LTL, LTTL, and LCL was significantly higher than that in the LTCL.

### 2.2. Dynamic Changes of Starch at Different Altitudes

During the decomposition observation period, the cumulative release of starch in the Chinese fir topsoil litter and canopy litter increased at four altitudes (Figure 2). The release rate of starch in Chinese fir topsoil litter and canopy litter during 0–120 days of decomposition was higher than that during 180–360 days. During the decomposition period of 120–300 days, the rank order of the cumulative release of starch in the LTL and LTTL groups at four altitudes was as follows: 620 m > 1410 m > 1003 m > 1968 m. During the decomposition period of 60–240 days, the release of starch in the LCL was as follows: 1003 m > 620 m > 1410 m > 1968 m. On day 300 of decomposition, the cumulative release percentage of starch in the LTTL and LTCL gradually stabilized. At the end of the observation, we observed that the cumulative release percentage of starch in the LTL was higher than that in the LTTL, and the cumulative release percentage of starch in the LCL was higher than that in the LTCL.

### 2.3. Dynamic Changes of NSC at Different Altitudes

The dynamics of NSC release in Chinese fir topsoil litter and canopy litter are similar (Figure 3). The release rate of NSC in Chinese fir topsoil litter and canopy litter during 0–180 days of decomposition was higher than that during 240–360 days. During the decomposition period of 0–360 days, the cumulative release of NSC in the LTL and LTTL was highest at 620 m and lowest at 1968 m. During the decomposition period of 120–240 days, the rank order of the cumulative release of NSC in the LCL group at four altitudes was as follows: 1003 m > 620 m > 1410 m > 1968 m. On day 300 of decomposition, the cumulative release percentage of NSC in the LTTL, LCL, and LTCL at four altitudes gradually stabilized. At the end of the test observation, the cumulative release percentage of NSC in the LTL, LTTL, and LCL was higher than that in the LTCL.

### 2.4. Interaction between Altitude and Decomposition Time on NSC Content 

The results of the ANOVA (Table 1) showed that altitude had significant effects on the NSC content in the LCL, LTL, and LTTL groups (*p* < 0.01) but had no significant effect on that in the LTCL group (*p* > 0.05). Decomposition time and the interaction between decomposition time and altitude had extremely significant effects on the NSC content of the Chinese fir topsoil litter and canopy litter (*p* < 0.01).

### 2.5. Correlation Analysis of the NSC and the C, N, and P Contents 

The correlation analysis between the NSC content and the carbon, nitrogen, and phosphorus contents in Chinese fir topsoil and canopy litter (Figure 4) revealed a significant positive correlation between NSC, starch, and nitrogen contents in canopy litter (*p* < 0.01); the soluble sugar and phosphorus content showed a significant positive correlation, whereas no significant correlation was observed between the NSC and carbon content (*p* > 0.05). In topsoil litter, the NSC, soluble sugar, and carbon content showed a significant positive correlation, while the starch content was not significantly correlated with carbon, nitrogen, or phosphorus content.

## 3. Discussion

### 3.1. Changes of NSCs Content in Trees at Different Altitudes

Our results verified the first hypothesis that the release of NSCs in Chinese fir litter show a decline trend with increasing of altitude. At the end of observation, the cumulative release percentage of NSC in Chinese fir topsoil and canopy litter were 33.26–46.43% at 620 m and 30.95–34.80% at 1968 m. The difference of the cumulative release percentage of soluble sugar between different altitudes is more significant than that of starch. This indicates that the release of starch was lower than that of soluble sugar under climate change conditions. The difference in the responses of starch and soluble sugars to climate change may result from their molecular structures. Soluble sugar, a monosaccharide solute that can directly involved in the physiological metabolism of plants, is easily leached by rainfall [17] and is significantly affected by external environmental factors [18]. Starch, as a reserve carbon source in forests, is a macromolecular polymer that is difficult to dissolve at normal atmospheric temperatures and is relatively insensitive to global warming. Chen et al. [19] simulated the effect of warming on the release of organic components in *Larix potaninii* litter using a top-open chamber and found that warming promoted the release of soluble sugars in the early stage of litter decomposition, which was similar to the results of our study. Additionally, warming not only alters the release rate of the NSC content in forest litter but also significantly accelerates the thermal degradation process of carbohydrates in tropical rain forest litter, generating a large amount of combustible gas and greatly increasing the possibility of forest fires [20].

The accumulation of NSC by trees is necessary for them to meet the energetic requirements of subsequent growth and metabolism when dealing with external threats and interference [21]. The storage of NSC is considered an important energy index for characterizing the survival ability of trees in response to stress. Studies have reported that NSCs contents in plants are closely related to the external environment [22,23]. Under high temperatures, the activities of enzymes involved in photosynthesis and respiration in plants are significantly increased [24], which enhances the growth and metabolism of plants and accelerates the consumption of soluble sugars in the body, leading to a significant decrease in the NSCs contents. The increase in global temperature not only altered the allocation strategy of NSCs in plants but also altered the release rate of easily decomposed components in the litter, thereby affecting the formation of forest soil organic matter [25]. A recent study showed that warming promotes the release of soluble sugars in litter [19], and the input of easily decomposed components in litter greatly increases the turnover rate of soil microorganisms. Dead residues generated after the completion of metabolic turnover of microorganisms are an important source of soil organic matter, accounting for over 50% of the total soil organic matter [25]. Therefore, warming may promote the formation of organic matter in forested soils and encourage the growth of plants.

### 3.2. Responses of the NSC Content in Different Types of Litter to Temperature Gradients

We partly confirmed the second hypothesis that the release of NSCs in the LCL is higher than that in the LTL. In contrast, the release of NSCs in the LTTL is higher than that in the LTCL. As an important part of the forest carbon pool, the change in NSC content not only affects the carbon storage of forests but also directly alters the litter decomposition process [26]. The availability of soil carbon sources significantly affects the intensity of soil microbial carbon metabolism during the decomposition of forest litter [5]. As the most easily decomposed carbon source in the early stage, NSC can significantly improve the metabolic turnover of soil microorganisms and accelerate the decomposition process of litter after it is leached into the soil by rainfall. Hu et al. [27] reported that the rainy season (from February to June) significantly accelerated the release and return of the NSC content in Chinese fir litterfall, and the NSC content of litter differed significantly among different organs, and the NSC content in leaf litter was higher than that in twig litter.

Our results showed that altitude had no significant effect on the NSC content in the LTCL group but had a significant effect on that in the LCL group. The release of NSCs in the LTL is higher than that in the LTTL, similar to Hu et al. [27]. Differences in the response of the NSC content in different tree organs to climate change are mainly caused by their physiological functions [28]. The leaf, as the main site of photosynthesis in forest trees, synthesizes a large number of carbohydrates to provide sufficient energy for the physiological metabolism of forest trees. To maintain the osmotic pressure of leaf cells and its high metabolic rate, the soluble sugar content in leaves is relatively high. Therefore, warming had a significant impact on the NSC content of plant leaves. Twigs are transport channels for carbohydrates in plants [29]. Under normal conditions, the NSC content of twigs is less responsive to the variational external environment [30], and most of the starch stored in the wood tissue is isolated [31]. 

The dead leaves and branches of Chinese fir constitute the topsoil litter and canopy litter. Zheng [32] reported that the quantity and nutrient accumulation of canopy litter accounts for an important proportion of the total amount of the dead leaves and branches, and significant differences were observed in the accumulation and distribution of nutrient elements between topsoil and canopy litter. In our study, we found that the NSCs contents in topsoil litter were significantly lower than that in canopy litter, and the difference in the NSCs contents between them may be related to their inconsistent decomposition times. Topsoil litter, as the product of canopy litter falling to the ground, experiences a longer atmospheric decomposition process. Studies have found that the high nutrient content in canopy litter can be considered to have evolved as a nutrient conservation strategy, which can maximize nutrient circulation within the tree and reduce the dependence on soil nutrients, especially in nutrient-deficient soil [16].

### 3.3. Correlation between the NSCs and the C, N, and P Contents

This study examined the soluble sugar, starch, C, N, and P contents in the Chinese fir topsoil and canopy litter. A correlation was observed between the NSC and nutrient contents. Nitrogen and phosphorus contents were the key factors that caused the release of NSCs in the canopy litter, whereas carbon content was the key factors causing the release of NSC and soluble sugar in the topsoil litter. The decomposition rate of forest litter was significantly affected by the C, N, and P contents. Previous research has shown that, when the C/N ratio is less than 25, P becomes the main limiting element for litter decomposition, whereas N becomes the main limiting element when the C/N ratio is greater than 25 [33]. Nitrogen is the key element affecting the growth and turnover of soil microbial communities, and its release rate significantly affects the loss of NSC in litter. In a warmer environment, acid phosphatase and catalase are more active in the soil, which accelerates the mass loss and microbial mineralization rate of dead leaves and branches [34], and further promotes the release and return of NSC. In the future, we should strengthen the long-term dynamic monitoring of the effects of climate change on forest litter with the help of remote sensing, unmanned aerial vehicles, and other scientific and technological means.

## 4. Materials and Methods

### 4.1. Study Site

This study was conducted at the Nature Reserve of Wuyi Mountain National Park (117°27′–117°51′ E, 27°33′–27°54′ N), Nanping City, Fujian Province, China. The region is characterized by a subtropical monsoon climate, with a mean annual temperature of 12–18 °C and a mean annual rainfall of 1486–2150 mm. Approximately 60% of the annual rainfall occurs during the main rainfall period is from March to June. The mean annual relative humidity is 83%, and the frost-free period is 253–270 days. The vertical climatic zones of the mountain are distinct, and the vertical distribution zone spectrum of vegetation is complete [35], which makes it a good research site for this study.

### 4.2. Test Materials Preparation

In August 2021, litter traps (1 × 1 m) were placed in a 15-year-old Chinese fir plantation at Xinkou National Forest Farm, Sanming City, Fujian Province, China. Twenty-seven litter traps were arranged in three 20 × 20 m plots evenly, and the distance between each plot is 20 m. The fresh Chinese fir topsoil litter was collected from each trap during September, with litter from other trees and understory vegetation discarded. Simultaneously, we measured tree diameters at breast height, tree height, and the height above ground of the highest dead branch for all trees in the plot, and three sample trees in each plot were selected for the sampling of dead leaves and branches in the canopy. A ruler was placed beside the tree trunk as a height marker, the canopy litter within the range of 2 m below the height of the highest dead branch were cut using branch clippers (Figure 5A), and nylon yarn nets were placed under each sample tree to collect canopy litter detached. The leaves and twigs (basal diameter ≤ 0.5 cm) of the Chinese fir topsoil and canopy litter were separated for natural air-drying and reserve use. Relatively intact branches (Figure 5B) of Chinese fir topsoil and canopy litter with a basal diameter of ≤0.5 cm were selected for separation and weighing and determined the mass ratio of twigs and leaves in standard branches about to be M_twig_:M_leaf_ = 1:3.

Traditional studies have paid less attention to the decomposition of canopy litter and a mixture of leaf and twig litter. In this study, four treatment groups of leaf topsoil litter (LTL), a mixture of leaf and twig topsoil litter (LTTL), leaf canopy litter (LCL), and a mixture of leaf and twig canopy litter (LTCL) were used. The mixture of leaves and twigs was bagged according to the mass ratio of M_twig_:M_leaf_ = 1:3, with each bag of leaves and twigs weighing 10 ± 0.005 g air dried (7.5 g leaves, 2.5 g twigs), and each bag of leaves weighing 10 ± 0.005 g. Four groups of samples were, respectively, placed into 20 × 20 cm nylon decomposing bags with a mesh size of 0.5 mm. A total of 144 bags of repeated samples were prepared for each group, and 576 bags were prepared for the four groups of experimental treatments.

### 4.3. Experimental Design

In September 2021, four altitudes (1968 m, 1410 m, 1003 m, and 620 m) were selected as the decomposition test sites in the nature reserve of Wuyi Mountain National Park (Figure 5C); the slope and aspect of the plots are basically identical at the four altitudes. Three 20 × 20 m plots were set at each altitude, and the distance between each plot is 20 m. Then, 0.5 kg of mixed soil samples at 0–20 cm deep were collected using an auger boring (QS–TZ, Qingyi Electronics, Tianjin, China. Diameter = 75 mm) in each plot using the five-point sampling method (the center and four corners 1.5 m away from the boundary). Automatic air and soil temperature and humidity monitors (TMS-4, Saifu Biotechnology, Shanghai, China) were installed at four altitudes. The soil temperature and humidity at 0–8 cm deep and atmospheric temperature of each altitude were recorded. Data were automatically recorded every 15 min. Basic information on the test sites at different altitudes is presented in Table 2.

Subsequently, the original topsoil litter in each plot were removed. Four groups of sample decomposition bags were placed on the ground surface at different altitudes, and used stainless steel U-shaped needle fixed the diagonal of the decomposition bag to prevent movement (Figure 5D). Twelve decomposition bags of the same treatment were arranged longitudinally, and a 50 cm interval zone was set between the different treatments. The decomposition of Chinese fir topsoil and canopy litter were conducted for 360 days, and sampling was conducted every 60 days. Two decomposition bags of the same treatment were retrieved in each plot as a duplicate sample and the sediment swept; six decomposition bags of each treatment group (three duplicate samples) were retrieved within each altitude, and 96 decomposition bags were retrieved every 60 days. The samples were dried to a constant weight at 65 °C, then ground into powder and sieved through a 0.178 mm mesh filter to determine the physical and chemical properties and the NSC content (Table 3). In this study, based on the work of Hoch et al. [36], NSC was defined as the sum of soluble sugars and starches and was determined using the phenol-–sulfuric acid method [37]. For each sample, the total C and N were measured using an Element Analyzer (VARIO MAX CN, Hanau, Germany), the total P was measured using the molybdenum–antimony colorimetric method with a 723A spectrophotometer (AOXI Instruments, Shanghai, China) [38], and the pH of the soil samples was measured using a pH meter (PHS-3C, Lei-ci, Shanghai, China).

### 4.4. Data Analysis

The mean value of NSCs released in the same treatment group were calculated using the SPSS 19.0 Statistics (SPSS Inc., Chicago, MI, USA) software. Differences of NSC contents in the same treatment group between different altitudes were determined using one-way ANOVA and post hoc least significant difference (LSD) tests. Topsoil and canopy litter responses were analyzed using a two-factor ANOVA, with altitude and decomposition time as fixed factors. Origin (version 2021) was used to draw the point line graphs and a heatmap of the correlation between the NSCs and the C, N, and P contents.

## 5. Conclusions

During the decomposition process of Chinese fir topsoil and canopy litter, NSCs had a high release at a low altitude. Our findings showed that the response mechanisms of the NSC content in different types of Chinese fir litter to altitude gradients were different. The release of NSCs in the mixture of leaf and twig topsoil litter is higher than that in the mixture of leaf and twig canopy litter. However, the response of NSC release in Chinese fir twig litter to altitude gradients is not well known and needs to be further investigated. The follow-up research should combine litter decomposition with soil to form a complete system. Taken together, the study is of great significance for a comprehensive understanding of the effect of climate change on Chinese fir litter decomposition.

## Figures and Tables

**Figure 1 plants-12-00729-f001:**
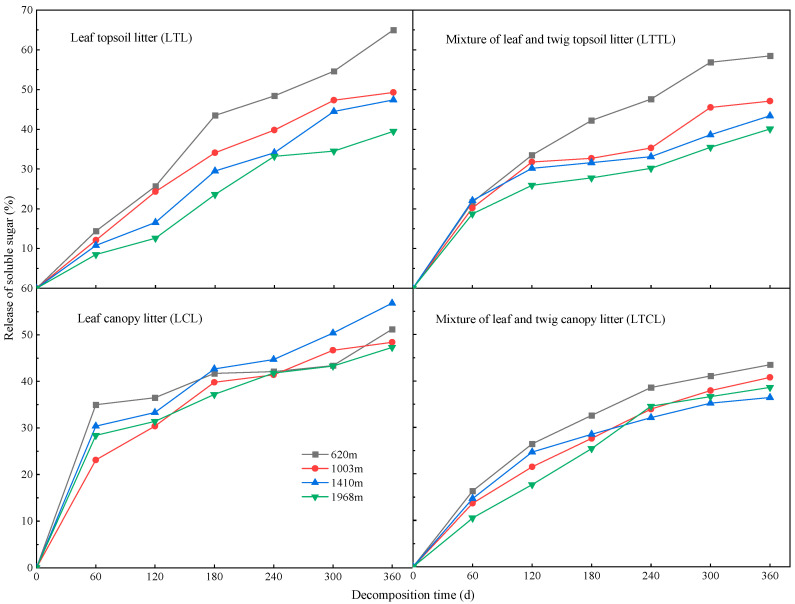
Cumulative release of soluble sugar in Chinese fir topsoil litter and canopy litter at different altitudes.

**Figure 2 plants-12-00729-f002:**
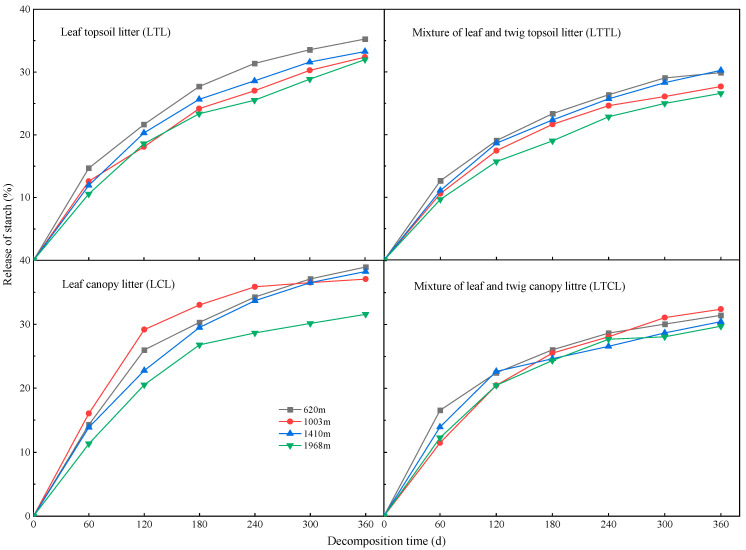
Cumulative release of starch in Chinese fir topsoil litter and canopy litter at different altitudes.

**Figure 3 plants-12-00729-f003:**
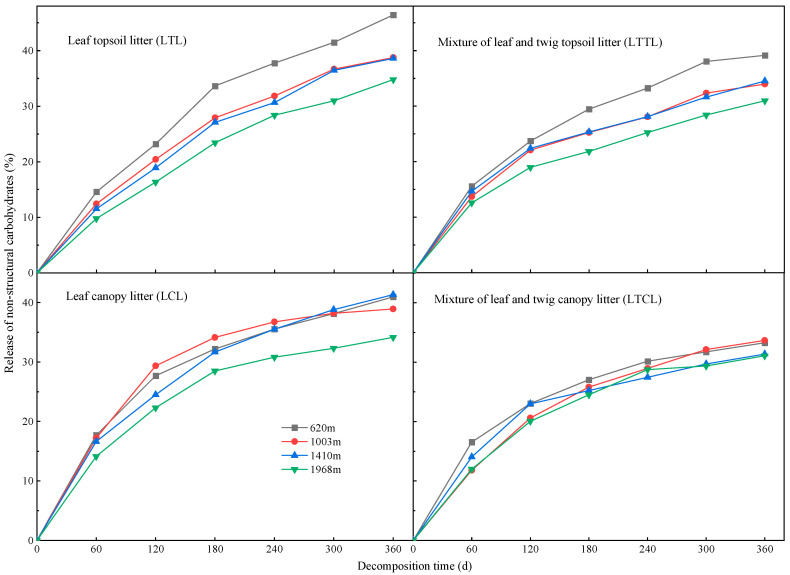
Cumulative release of NSC in Chinese fir topsoil litter and canopy litter at different altitudes.

**Figure 4 plants-12-00729-f004:**
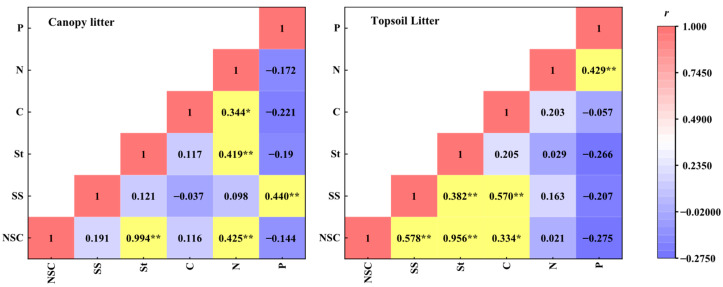
Correlation between non-structural carbohydrates (NSCs) and carbon (C), nitrogen (N), and phosphorus (P) contents in topsoil and canopy litter of Chinese fir. Soluble sugars (SS); starch (St). The value is correlation coefficient, asterisks denote significant correlations (** *p* < 0.01, * *p* < 0.05).

**Figure 5 plants-12-00729-f005:**
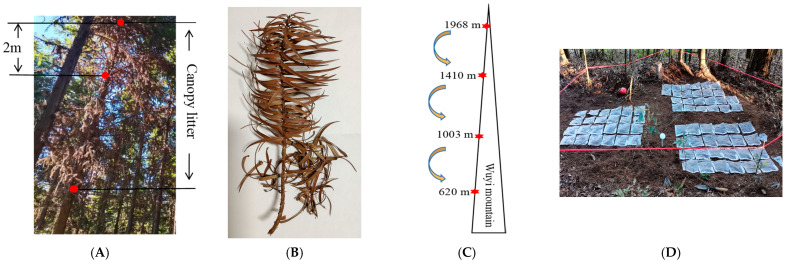
Material collection and experiment design. (**A**) Canopy litter collection range. (**B**) Relatively intact branches. (**C**) Locations of the decomposition sites. (**D**) Placement of decomposition bags.

**Table 1 plants-12-00729-t001:** Two-way ANOVA of the effects of altitude and decomposition time on NSC content in Chinese fir topsoil litter and canopy litter.

Treatment	Source of Variation	SS	*df*	F	*p*
Leaf topsoil litter (LTL)	Time	11.22	5	15.77	<0.001
	Altitude	4.63	3	10.84	<0.001
	Time × Altitude	21.36	15	10.01	<0.001
Mixture of leaf and twig topsoil litter (LTTL)	Time	66.16	5	75.50	<0.001
	Altitude	4.06	3	7.71	<0.001
	Time × Altitude	18.52	15	7.04	<0.001
Leaf canopy litter (LCL)	Time	46.88	5	20.80	<0.001
	Altitude	12.70	3	9.39	<0.001
	Time × Altitude	39.43	15	5.83	<0.001
Mixture of leaf and twig canopy litter (LTCL)	Time	26.75	5	13.91	<0.001
	Altitude	2.65	3	2.30	0.089
	Time × Altitude	43.68	15	7.57	<0.001

**Table 2 plants-12-00729-t002:** Basic information of different altitudes gradient test sites.

Altitude/m	Average Annual Air Temperature/°C	Average Annual Soil Temperature/°C	Average Annual Soil Humidity/%	Soil Total Nitrogen (g·kg^−1^)	Soil Total Phosphorus/ (g·kg^−1^)	Soil pH
692	16.9	17.9	14.2	0.43 ± 0.07	0.32 ± 0.01	4.76
1003	15.0	14.8	16.8	0.77 ± 0.12	0.30 ± 0.01	4.41
1410	12.9	12.7	18.7	0.71 ± 0.13	0.46 ± 0.06	4.38
1968	10.2	11.5	21.4	0.54 ± 0.04	0.30 ± 0.03	4.59

Annual average air temperature, annual average soil temperature and annual average soil humidity are from September 2021 to September 2022.

**Table 3 plants-12-00729-t003:** Initial physical and chemical properties of Chinese fir topsoil and canopy litter.

Treatment	Soluble Sugar Content/%	Starch Content/%	NSC Content/%	Total Carbon/(g·kg^−1^)	Total Nitrogen/(g·kg^−1^)	Total Phosphorus/(g·kg^−1^)
LTL	1.56 ± 0.03 A	2.59 ± 0.08 C	4.15 ± 0.11 C	506.03 ± 20.54 A	15.22 ± 0.20 C	0.35 ± 0.11 A
LTTL	1.54 ± 0.04 A	3.22 ± 0.32 D	4.76 ± 0.29 BC	503.69 ± 13.62 A	13.98 ± 0.22 D	0.29 ± 0.07 A
LCL	1.34 ± 0.19 B	6.73 ± 0.28 A	8.07 ± 0.68 A	496.42 ± 16.53 A	17.60 ± 0.31 A	0.34 ± 0.02 A
LTCL	0.80 ± 0.04 C	4.46 ± 0.25 B	5.26 ± 0.25 B	494.50 ± 14.96 A	16.16 ± 0.22 B	0.25 ± 0.02 A

LTL: leaf topsoil litter; LTTL: mixture of leaf and twig topsoil litter; LCL: leaf canopy litter; LTCL: mixture of leaf and twig canopy litter. Different uppercase letters denote significant differences in Initial physical and chemical properties among treatment groups (*p* < 0.05).

## Data Availability

The data presented in this study are available on request from the corresponding author. The data are not publicly available due to privacy.

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
