# Peer review of "Dynamics of Non-Structural Carbohydrates Release in Chinese Fir Topsoil and Canopy Litter at Different Altitudes"

_plants, 2023, doi:10.3390/plants12040729_

Round 1

Reviewer 1 Report

The current manuscript presents an interesting study of litter decomposition dynamics in a fir woodland in China. I believe the results would be of interest to those working on nutrient cycling under changing climatic conditions in similar woodland environments and should be of interest to the wider readership of the journal. However, I believe the authors need to address several issues in the manuscript before it can be considered for publication.

My main concern is with the lack of details in the methodological approach. I missed some information to enable me to judge the suitability of the experimental design and data analyses. For instance, how many litter traps were used in each plot? How far apart were the three plots located? The placement of the decomposition bags along an altitudinal gradient is not clear either (a map depicting the locations of the decomposition sites would be helpful; the schematic diagram in Fig. 4 is inadequate and the photo is barely visible). These questions are pertinent for the choice of numerical analyses as the authors could be incurring pseudoreplication. The authors stated they used ANOVA to test the differences between treatments, but why was independence of errors assumed between treatments? Were the samples from the same plot averaged or treated as separate sample units? Would a Mixed Effects Model have been more appropriate? As mentioned previously, there is not enough information in the Methods section to help the reader judge the methodological choices.

In addition, there is no explanation as to how ‘canopy litter’ was identified. Not all readers will be familiar with this type of work, and it would be helpful if the authors could explain how canopy litter (if I understood correctly, litter that is still attached to the canopy) was distinguished from live canopy (i.e., were the leaves wilted? Discoloured? Was there a cut-off point the authors used to separate dead from live canopy?). This needs a brief clarification. I was also somewhat confused by the use of the term ‘aboveground litter’ to distinguish between canopy litter and litter collected from litterfall. Isn’t canopy litter also aboveground litter? How about referring to litterfall as ‘topsoil litter’?

Moreover, the authors give the impression they manipulated changes in temperature somehow (P9, L326), though it is more likely this sentence is poorly written, and the authors meant to say changes in altitude served as a proxy for changes in temperature. Please rewrite for clarity. Also, what does ‘based on the principle of consistent site conditions’ (P9, L323) mean? I cannot understand this sentence and have not heard of such principle before (is there a reference?).

As a minor point, I’m not convinced the term ‘global warming’ is the best for the title and the text, especially considering the term now commonly used is ‘climate change’ (not all parts of the globe are warming). I’d suggest the authors consider changing the terminology.

The Abstract is not summarising the methods and results well enough and, in my opinion, comes across as misleading at points (e.g., P1, L18-19 and L21-22). Some more clarity as to what was done (methods, including where the study took place) is needed. In addition, I do not think the authors can make sweeping generalisations to the effects of ‘global warming’ since, if I understood correctly, they did not manipulate changes in temperature per se, but used changes in altitude as a proxy for temperature.

I felt the Introduction could motivate the study better. The first paragraph reads more like a discussion (and some of it like methods) and does not draw attention as to why this study is pertinent (the first few sentences are too broad and vague in my opinion). I’d suggest starting with an overview of the importance of litter decomposition rates and NSC in forest ecosystems to then make the link to climate change (what changes have been recorded so far as a result of climate change? What changes are expected?). The knowledge gaps could then be highlighted before setting out the objectives of the study. I also thought this section was a little thin on evidence and I missed some references in places (e.g., P1, L34-37).

The Results section seems overly long and detailed, while some of the figures are not referenced in the text (e.g., Figs 1 and 2). I’d suggest letting the figures speak for themselves and only present the main message in the text (which can be discussed later in more detail in the Discussion section). I would also suggest the authors consider presenting percentage loss in figures 1 and 2 instead of percentage content. I wonder if that would provide a better comparison between treatments and allow readers to better visualise the effects of altitude and time on decomposition. In addition, I wondered why section 2.3 is illustrate with a table and not a line plot like the previous sections. I’d suggest keeping consistent (graphs are better to visualise than tables). Could the correlations between variables be presented as a series of bivariate scatterplots instead of a correlation matrix? The numerical results of the correlations could then be presented on each panel. I wonder if that would make the relationship between variables easier to visualise.

The Conclusions read like a repetition of the Results section and seem unnecessary, not to mention the final sentence is extremely vague and does not seem grounded in evidence. I’d suggest removing it or rewriting it. For instance, how can this type of research be further developed? What were the limitations of the methods used in this study? What knowledge gaps should be addressed next to complement the results presented in this manuscript? I believe these would be more interesting points to discuss in the concluding remarks.

Finally, the text could benefit from some English language editing throughout for clarity.

Author Response

Dear reviewer, thank you for your suggestions to make the article better. The details of revisions please see the attachment.

Reviewer 2 Report

The manuscript subject corresponds to the journal. The manuscript can be accepted for publication without significant changes. However, I would like to reccomend add hypothesis to the Introduction.

Author Response

Dear reviewer, 

we are very grateful to the critical comments and thoughtful suggestions from reviewer and editor. Based on these comments and suggestions, we have carefully modified manuscript according to our best knowledge. All comments are addressed below in the revised manuscript. The line numbers referred to below are in the revised manuscript.

Point 1: The manuscript subject corresponds to the journal. The manuscript can be accepted for publication without significant changes. However, I would like to reccomend add hypothesis to the Introduction.

Response 1: Dear reviewer, thank you for your suggestions to make the article better. In the introduction, we added the hypotheses: We hypothesized that (1) due to the NSCs contents in plants is closely related to the temperature, the release of NSCs in Chinese fir litter show a decline trend with increasing of altitude; (2) the release of NSCs in Chinese fir canopy litter is higher than that in the topsoil litter (P2, L74-76). Our results verified the first hypothesis that the release of NSCs in Chinese fir litter show a decline trend with increasing of altitude. At the end of observation, the cumulative release percentage of NSC in Chinese fir topsoil and canopy litter were 33.26%–46.43% at 620m and 30.95%–34.80% at 1968m (P6, L153-156). And we partly confirmed the second hypothesis that the release of NSCs in the leaf canopy litter is higher than that in the leaf topsoil litter. In contrast, the release of NSCs in the mixture of leaf and twig topsoil litter is higher than that in the mixture of leaf and twig canopy litter (P7, L190-192).
